# Change of Internet Use and Bedtime among Junior High School Students after Long-Term School Closure Due to the Coronavirus Disease 2019 Pandemic

**DOI:** 10.3390/children8060480

**Published:** 2021-06-07

**Authors:** Hideki Nakayama, Takanobu Matsuzaki, Satoko Mihara, Takashi Kitayuguchi, Susumu Higuchi

**Affiliations:** 1Hokujinkai Asahiyama Hospital, Sapporo 064-0946, Hokkaido, Japan; 2National Hospital Organization Kurihama Medical and Addiction Center, Yokosuka 239-0841, Kanagawa, Japan; takanobum@gmail.com (T.M.); mihara.satoko.pd@mail.hosp.go.jp (S.M.); kitayuguchi_kurihama@yahoo.co.jp (T.K.); h-susumu@db3.so-net.ne.jp (S.H.)

**Keywords:** internet addiction, smartphone, Young’s Diagnostic Questionnaire, bedtime, COVID-19

## Abstract

Most schools in Japan were closed in spring 2020 due to the novel coronavirus disease 2019 (COVID-19) pandemic. We investigated lifestyle and internet use among junior high school students across eight schools after long-term school closure and compared the data with those we obtained from previous surveys. In the summers of 2018, 2019, and 2020, we conducted questionnaire surveys on seventh-grade students from the same schools. In total, 2270 participants were analyzed. All questionnaires included items regarding background, bedtime, and internet use. The participants of the 2020 survey had significantly less sleepiness during classes and longer internet use times compared with those of the previous surveys. In the 2020 survey, the rate of problematic internet use (Young’s Diagnostic Questionnaire score, ≥5) was not significantly different from the results of previous surveys. The COVID-19 pandemic might have strongly influenced the sleepiness experienced by students in classes and increased the time spent using the internet since the summer of 2020. Our results indicate the need for attempts to encourage students to improve their sleep habits and moderate their media use.

## 1. Introduction

Novel coronavirus disease 2019 (COVID-19) infections first occurred in Wuhan City, China, in December 2019 and subsequently spread rapidly around the world. The World Health Organization declared a global health emergency on January 30, 2020 [1]. In spring 2020, the Japanese government recommended that people stay at home; most schools and facilities were closed, and most events were canceled. As a result, many people may have considered the internet to be increasingly indispensable as a tool for communication, education, entertainment, information seeking, and other similar activities.

On the other hand, internet addiction (IA) or problematic internet use (PIU) has become a significant problem among the youth. A Japanese school-based questionnaire survey revealed that 7.9% of junior and senior high school students were suspected of having IA [2]. Studies have also reported that IA among adolescents was strongly associated with worsening psychiatric symptoms [3], sleeping disturbances [4,5], later bedtimes [5,6], obesity [7], and poor academic achievement [8].

However, an official diagnostic criterion for IA does not exist. Internet use includes such activities as gaming, social networking, watching movies, and information seeking. For example, Baggio et al. [9] reported that smartphone addiction, gaming addiction, and cybersex addiction were relatively independent constructs, but these behavioral addictions were often related to IA. This finding suggests that IA is the aggregation of various types of internet-related addictions. Nevertheless, as almost all adolescents used some internet services, we considered that most adolescents have a better understanding of IA than of specific addictions (e.g., gaming addiction, social networking service addiction, and movie addiction).

Some experts have expressed concern that the lifestyle changes brought about by the COVID-19 pandemic may cause some vulnerable individuals to develop problematic psychoactive substance use and other reinforcing behaviors, such as gambling, video gaming, movie watching, using social media, and internet surfing [10,11]. For example, Sun et al. [12] found that 46.8% of their online questionnaire survey participants reported increased dependence on internet use and that 16.6% of them reported longer internet use time during the COVID-19 pandemic. In addition, some research has shown that people’s sleep situation has changed during the COVID-19 pandemic. For example, Zhang et al. [13] reported that the COVID-19 death count had a direct negative impact on college students’ general sleep quality. Pieh et al. [14] reported that 18.6% of young adults were suspected of having insomnia during lockdown. Lee et al. [15] reported that residents in the United States and European countries delayed their bedtime and slept longer than usual during the COVID-19 pandemic.

However, reported data on changes in adolescents’ lifestyle and internet use after long-term school closure due to the COVID-19 pandemic are few. We thus investigated the lifestyle and internet use of public junior high school students after long-term school closure due to the pandemic and compared the data with those we obtained from previous surveys to further examine the impact of this pandemic.

## 2. Materials and Methods

### 2.1. Situation before and after School Closure Due to the COVID-19 Pandemic in City A

In Japan, COVID-19 infections started to first occur in January 2020 and gradually increased. The Japanese government requested that all elementary and high schools be temporarily closed from 2 March [1]; all public junior high schools in city A, where our surveys were conducted, were closed from 3 March to 31 May (only opening and entrance ceremonies were held on 6 April). In addition, from 7 April to 25 May, the Japanese government declared a state of emergency in Kanagawa Prefecture, where city A is located [1]. During that period, many public and private facilities were closed, most events were canceled, and people were recommended to stay at home. However, in 2020, there was no city-wide lockdown and curfew order because of the COVID-19 pandemic in Japan. All public junior high schools in city A reopened from 1 June, and shortened class hours continued until 13 July. The summer vacation period was from 1 August to 23 August, which was shorter than the usual in previous years. Extracurricular activities in schools were suspended until 28 June and restarted from 29 June with reduced activity time.

### 2.2. Participants

This study consisted of three parts: Survey I was conducted in June 2018, survey II was conducted in June 2019, and survey III was conducted in July 2020. All surveys were administered in the same eight public junior high schools in city A, Kanagawa Prefecture, Japan. Kanagawa Prefecture is located next to Tokyo, the capital of Japan. The population of city A is approximately 200,000. All participants enrolled in each survey were seventh-grade students. Surveys I, II, and III targeted 868, 952, and 1048 participants, respectively. A total of 814 students participated in survey I, of whom 734 completed all the items analyzed in this study; a total of 871 students participated in survey II, of whom 734 completed all the items analyzed in this study; and a total of 961 students participated in survey III, of whom 802 completed all the items analyzed in this study. Overall, we analyzed 2270 participants.

### 2.3. Procedure and Assessment

Before the surveys were conducted, we and the teachers sent the students’ guardians a letter explaining the purpose of the study, the main contents of the questionnaire surveys, the opportunity to refuse participation, and the protection of each student’s personal information. At the beginning of each questionnaire, we explained the purpose of the study, main contents of the questionnaire, opportunity to submit a blank sheet to refuse participation in the survey, protection of each student’s personal information, and the presentation at a conference and the publication of this study’s findings. The teachers also explained the same contents to the students. Students thereafter filled out the anonymous self-report questionnaire at their respective classrooms and submitted it to their teachers.

In this study, we analyzed items regarding age, sex, bedtimes on weekdays (“What was the average time you went to bed on the schooldays, excluding days with extracurricular activities only, in the previous 30 days?”), bedtimes on holidays (“What was the average time you went to bed on the non-schooldays, including days with extracurricular activities only, in the previous 30 days?”), sleepiness during classes, frequency of participation in lessons or crammers (measured only in surveys II and III), type of internet service used, type of internet device used, smartphone ownership, and average durations of daily internet use for purposes other than study in the last month on weekdays and holidays. Questions other than those that sought to determine the average duration of daily internet use were in multiple-choice format. We also administered the Japanese version of Young’s Diagnostic Questionnaire (YDQ) [16,17]. The YDQ is a scale for IA and comprises eight yes/no questions with scores ranging from 0 to 8. As proposed by Young [16], respondents with YDQ scores ≥5 were classified as addictive internet users. The Japanese version of the YDQ [17] was back-translated to avoid language bias. Participants whose YDQ scores ranged from 0 to 4 were classified as the normal internet use (NIU) group, whereas those whose YDQ scores ranged from 5 to 8 were classified as the PIU group.

### 2.4. Statistical Analysis

Table 1 and Table 2 show the comparison of the categorical data (e.g., age, sex, bedtime, sleepiness during classes, frequency of participation in lessons or crammers, smartphone owner, type of internet device used, type of internet service used, and PIU) for each survey year performed using *χ*^2^ analysis. A comparison of continuous data (e.g., average duration of daily internet use and average YDQ score) for each year was done using one-way analysis of variance (ANOVA). Figure 1 and Figure 2 show a comparison of internet use (in minutes) on weekdays and holidays among groups as per each bedtime, as assessed using ANOVA. A post hoc analysis between the group with bedtime before or at 21:59 and groups with bedtime at or after 22:00 (22:00–22:59, 23:00–23:59, and at or after 0:00) was performed using Dunnett’s test. Figure 3 shows the comparison of degree of sleepiness during classes as per bedtime between each survey, as assessed using χ2 analysis. For each bedtime, a comparison of the internet use duration (in minutes) on weekdays among groups as per the survey year was performed using ANOVA. A post hoc analysis between the group from survey III and those from the other surveys (surveys I and II) was performed using Dunnett’s test. Table 3 and Table 4 show a comparison of the categorical data (e.g., age, sex, bedtime, sleepiness during classes, frequency of participation in lessons or crammers, smartphone owner, type of internet device used, and type of internet service used) between the NIU and PIU groups, as assessed using *χ*^2^ analysis. A comparison of continuous data (average duration of daily internet use) between the NIU group and the PIU group was made using *t*-test. Figure 4 shows a comparison of the duration of internet use (in minutes) on weekdays and holidays among groups for each survey year, as assessed using ANOVA. A post hoc analysis between the group from survey III and those from the other surveys (surveys I and II) was performed using Dunnett’s test. Table 5 shows the results of the multivariate logistic regression analysis that was conducted to determine the variables associated with PIU. Independent variables included sex, age, survey years, bedtime on weekdays and holidays, type of internet devices used, and type of internet services used.

A *p* value of 0.05 was considered statistically significant. All statistical analyses were performed using SPSS 27.01.

## 3. Results

### 3.1. Comparison of Characteristics, Lifestyle, and Internet Use of Participants from Each Survey

The results of comparisons of age, sex, bedtime, and frequency of participation in lessons or cramming between the surveys are shown in Table 1. The rate of 13-year-old participants in survey III was higher than the rates of such participants in the other surveys. In all surveys, 7.6% to 10.9% of the participants went to sleep at or after 12:00 AM on weekdays, and 12.1% to 15.3% went to sleep at or after 12:00 AM on holidays. The rate of participants who went to bed after 12:00 AM on weekdays was higher in survey III than in the other surveys.

In all the surveys, 21.7% to 25.9% of the participants answered that they were “common” or “always” sleepy during classes. The rate of participants in survey III who answered “common” or “always” was lower than that of participants in the other surveys. The frequency of participation in lessons or crammers tended to be lower in survey III than in survey II.

The results of comparisons of smartphone ownership, type of internet device used, type of internet service used, average duration of daily internet use, and YDQ score between the surveys are shown in Table 2. The Cronbach’s *α* of the YDQ was 0.664 in this study. The rate of smartphone owners in survey III was higher than the rates of smartphone owners in the other surveys. The availability of smartphones and portable game consoles gradually increased, whereas that of feature phones gradually decreased. The availability of social networking services as well as that of movie and/or music services gradually increased. The average durations of daily internet use on weekdays and holidays in survey III were longer than those in the other surveys. The average YDQ score and rate of PIU (YDQ score, ≥5) did not significantly differ between the surveys.

The results of comparisons of average minutes of daily internet use by bedtime on weekdays between the surveys are shown in Figure 1. In all surveys, the participants who went to bed later on weekdays had longer average internet use time. The average internet use time in survey III was longer than the averages in the other surveys at all bedtime hours.

The results of comparisons of average minutes of daily internet use by bedtime on holidays between the surveys are shown in Figure 2. In all surveys, the participants who went to bed later on holidays had longer average internet use times. The average internet use time in survey III was longer than the averages in the other surveys at all bedtime hours.

The results of comparisons of sleepiness during classes and bedtime on weekdays between the surveys are shown in Figure 3. The rate of participants in survey III who answered “common” or “always” sleepy during classes tended to be lower than the rates of participants in the other surveys who answered the same. The average duration of daily internet use by bedtime on weekdays in survey III was longer than the average durations in the other surveys.

### 3.2. Comparison of Characteristics, Lifestyle, and Internet Use of Participants Between the NIU and PIU Groups

The results of comparisons of age, sex, bedtime, and frequency of participation in lessons or crammers between the NIU and PIU groups are shown in Table 3. Bedtimes on weekdays and holidays in the PIU group were significantly later than those in the NIU group. The rate of sleepiness during classes in the PIU group was significantly higher than that in the NIU group.

The results of comparisons of smartphone ownership, type of internet device used, type of internet service used, and average duration of daily internet use between the NIU and PIU groups are shown in Table 4. The rate of smartphone owners in the PIU group was significantly higher than that in the NIU group. The rates of “tablet,” “portable game console,” and “stationary game console,” users in the PIU group were significantly higher than those in the NIU group. The rate of users of contents other than “Information and news searching” in the PIU group was significantly higher than that in the NIU group. The average durations of daily internet use on weekdays and holidays in the PIU group were significantly longer than those in the NIU group.

The results of comparisons of average minutes of daily internet use on weekdays and holidays by the NIU and PIU groups between the surveys are shown in Figure 4. In the NIU group, the average minutes of daily internet use on weekdays and holidays in survey III were significantly longer than those in surveys I and II. In the PIU group, the average durations of daily internet use on weekdays and holidays were not significantly different between the surveys.

Table 5 shows the association between multiple factors (sex, age, survey years, bedtime on weekdays and holidays, type of internet device used, and type of internet service used) and the risk of PIU. This logistic regression model was statistically significant (*p* < 0.001), and the Nagelkerke *R*^2^ value was 0.200; “bedtime at or after 12:00 PM on holidays” (odds ratio: 2.949), “portable game console user” (odds ratio: 2.652), “e-mail, chat, and internet telephone user” (odd ratio: 2.289), “blog and internet bulletin board user” (odds ratio: 2.217), and “shopping and auction user” (odds ratio: 1.832) were associated with a higher likelihood of PIU.

## 4. Discussion

We compared the results of our recent survey with those of our previous surveys to evaluate the changes in lifestyle and internet use among junior high school students after long-term school closure due to the COVID-19 pandemic.

Later bedtimes and shortened sleep times are well known to be strongly associated with health and social problems, longer screen time, and PIU. For example, Hysing et al. [18] reported that adolescents who go to bed between 10:00 and 11:00 PM have the best grade point averages, whereas those who go to bed after 11:00 PM have lower grade point averages. Olds et al. [19] reported that later bedtime is correlated with longer screen time and less moderate vigorous physical activity. Kohyama et al. [20] reported that reduction of average sleep duration was associated with longer screen time on weekdays and longer time spent on after-school activities in junior high school students. In addition, studies have shown that PIU or addictive internet use is correlated with poor sleep quality, daytime sleepiness, shortened duration of night sleep [5,21], and later bedtime [5,6].

In our study, later bedtime was strongly correlated with longer internet use time and PIU. The participants who went to bed at or after 12:00 AM on weekdays spent an average of 3–4 h on internet use. Especially on weekdays, these 3–4 h occupied large proportion in their life. Longer internet time might lead to later bedtime and shortened sleep time. In all surveys, the participants who went to bed at or after 12:00 AM on weekdays and holidays spent more time using the internet by an average of 1 h or longer compared with those who went to bed before or by 11:59 PM. If students who go to bed later would spend less time on the internet, they may be able to go to bed earlier.

Some researchers reported that adolescents’ sleep time was extended [22], sleep quality improved [23,24], and bedtime and wake-up time [24] were delayed during school closure due to the COVID-19 pandemic. For example, Drugun et al. [25] reported that the median of sleep duration on weekdays, the rate of students who felt refreshed after waking up on weekdays, and the median of computer/tablet use time were increased during lockdown due to the COVID-19 pandemic in secondary school and medical school students.

In our study, the proportion of participants in survey III who answered “common” or “always” sleepy during classes showed a tendency to be lower than that of participants in the other surveys, although their average duration of daily internet use was longer. The participants in survey III exhibited a lower frequency of participation in lessons or cramming compared with those in survey II. As school club activities and most events were restricted and school time was shortened during the period when survey III was conducted, students might have been less physically tired than in the previous prepandemic years. Furthermore, these factors might have affected the participant’s bedtime and wake-up time in survey III. The students’ sleepiness during classes might be related not only to bedtime and internet use but also to the frequency of their participation in lessons or crammers, club activities, events, outdoor activities, amount of exercise, and other activities.

For example, Tokiya et al. [26] reported that shorter school sport or club activity hours related to sleep disturbance in their boys’ high school students; however, extracurricular learning hours were not associated with sleep disturbance in boys and girls. Nemoto et al. [27] reported that busy life style was associated with poor sleep quality in their elementary and junior high school students. In our study, the generous schedule might have exerted a positive impact on the students’ sleep and decreased sleepiness during classes in survey III. However, as we did not investigate sleep and other factors (e.g., caffeine or alcohol intake and wake-up time) in each survey, we could not identify any causal relationship that could have contributed to sleepiness during classes. Comprehensive measures, including prevention of PIU, reconsideration of the students’ schedule, and promotion of early bedtime, are needed to improve sleep quality in adolescents.

Some studies have reported on the changes in internet use over the years. For example, the Cabinet Office of the Government of Japan [28] reported that the availability of total internet services was 79.4% in 2014 and 95.1% in 2019, and the average durations of internet use of Japanese junior high school students were 138.3 min in 2016, 148.7 min in 2017, 163.9 min in 2018, and 176.1 min in 2019, showing a tendency to gradually increase. In the same survey, the availability of smartphones and tablets in junior high school students substantially increased from 2014 to 2019. On the other hand, the average duration of television viewing time of Japanese teens was reported to be 102.5 min in 2013, 95.8 min in 2015, and 73.3 min in 2017, showing a tendency to gradually decrease [29]. The tendency of internet use time to gradually increase may have been associated with the increasing availability of internet devices and decreasing television viewing time.

Our results also showed that the average internet use time increased annually, and this tendency was remarkable in the NIU group. In addition, the average duration of internet use in survey III was considerably longer that the average durations in the other surveys. Some research has also found changes in the prevalence of PIU and/or addictive internet use over the years. For example, Kojima et al. [30] reported that the prevalence of PIU (YDQ score, ≥40) in junior high school students did not significantly change over their three-year survey period. On the other hand, Kinjyo and Osaki [31] reported that the prevalence of addictive internet use (YDQ score, ≥5) increased by approximately 6% between 2012 and 2017 among junior high school students. Kawabe et al. [32] reported that the average Young’s IA Test score and prevalence of addictive internet use (Young’s IA Test score, ≥70) significantly increased from 2.0% in 2014 to 4.3% in 2018 among junior high school students. In our study, we found that the average YDQ score and rate of PIU in survey III increased and were higher than those in surveys I and II, but the differences were not statistically significant.

The increased rate of smartphone owners, decreased frequency of participation in lessons or cramming, shortened class hours and extracurricular activities, canceled events, long-term school closure, an annually increasing trend in the duration of internet use, and other related outcomes of the COVID-19 pandemic might have affected the significant increase in internet use time in survey III. Moreover, these factors might have, to some extent, influenced PIU among the participants in survey III, although the effect was not significant. At the time survey III was conducted, it had been 4 or 5 months since the COVID-19 pandemic began to spread in Japan, and this period might not have been long enough for the students to exhibit significantly increased PIU. However, previous studies have shown that excessive internet use was associated with poor mental health state [33], shortened sleep time [34], and addictive internet use [2,6]. In our study, logistic regression analysis showed that PIU was associated with “bedtime at or after 12:00 PM on holidays”, “portable game console used”, “e-mail, chat, and internet telephone used”, “blog and internet bulletin board used”, and “shopping and auction used.” These and other related factors warrant close monitoring for the prevention of excessive internet use and PIU.

The number of people in Japan infected with COVID-19 increased rapidly again from the end of 2020, and the Japanese government declared another state of emergency in several prefectures, including Kanagawa, on 7 January 2021. Most in-school classes in Japan have been normalizing with consideration for the prevention of COVID-19 infection, but extracurricular activities and many events have been either canceled or scaled down. Especially during the COVID-19 pandemic, the internet could be more useful for many adolescents as a tool for communication, education, entertainment, information seeking, and other similar activities. However, excessive internet use and PIU are not favorable outcomes for them. Opportunities for adolescents to engage in various activities (e.g., sports, arts, and outdoor play) safely instead of using electronic devices are needed. Education and support to encourage proper media use are also crucial.

### Limitations of the Study

This study has several limitations. First, our assessment was based solely on the students’ self-reported data, which might have been affected by inaccurate memories and cognitive distortions. Second, the characteristics and backgrounds of the participants across all surveys did not always match. However, because all surveys were conducted at the same public junior high schools, we assumed homogeneity of the participants to a certain degree. Third, some PIU risk factors (e.g., psychiatric symptoms, developmental disorders, socioeconomic status, and family function) were not considered in this study owing to the limited number of questions. Fourth, given that only seventh-grade students were included, the results may not be generalizable to other students; surveys on students from other grade levels and adults may provide different results. Fifth, considering the limited number of questions in our survey, some sleep factors (e.g., waking time, sleep qualities, sleep time, and sleepiness scale score) were not included. Future studies should address these concerns.

## 5. Conclusions

In the summers of 2018, 2019, and 2020 (after school closure due to the COVID-19 pandemic), questionnaire surveys among seventh-grade students were conducted. The participants in the 2020 survey had significantly later bedtimes and longer internet use time than those in previous surveys. However, in the 2020 survey, the rates of PIU were not significantly different from the results of previous surveys. Excessive internet use and PIU are unfavorable outcomes for adolescents. Education and support to encourage proper media use and appropriate sleep habits are also crucial.

## Figures and Tables

**Figure 1 children-08-00480-f001:**
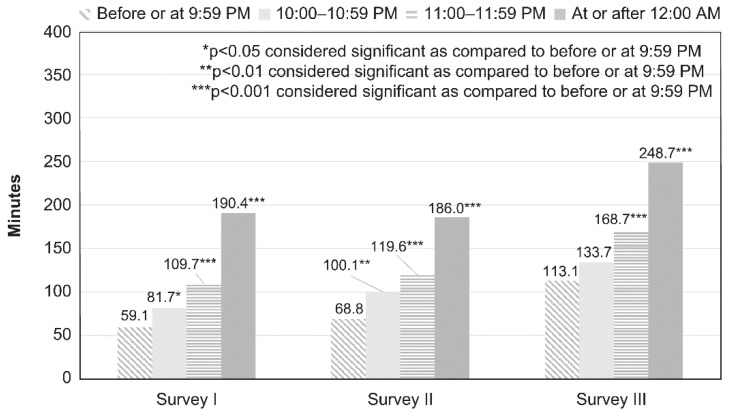
Average daily Internet time (minutes) on weekdays according to bedtime on weekdays in each survey. A comparison of the duration of internet use (in minutes) on weekdays among the groups as per bedtime, as assessed using ANOVA. A post hoc analysis between the group with bedtime before or at 9:59 PM and groups with bedtime at or after 10:00 PM (10:00–10:59 PM, 11:00–11:59 PM and at or after 12:00 PM) was performed using Dunnett’s test.

**Figure 2 children-08-00480-f002:**
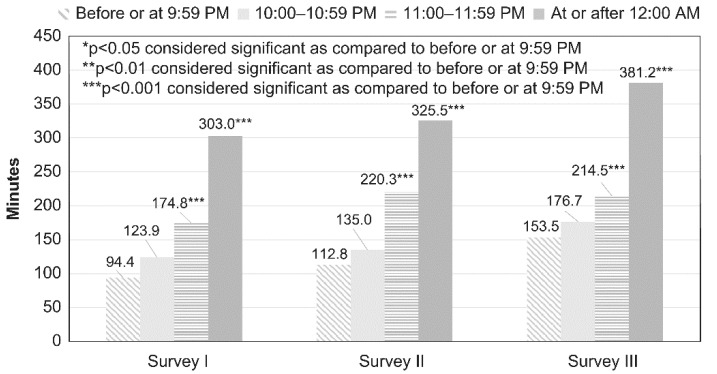
Average daily Internet minutes on holidays by bedtime on holidays in each survey. A comparison of the duration of internet use (in minutes) on holidays among groups as per bedtime was performed using ANOVA. A post hoc analysis between the group with bedtime before or at 9:59 PM and groups with bedtime at or after 10:00 PM (10:00–10:59 PM, 11:00–11:59 PM and at or after 12:00 PM) was performed using Dunnett’s test.

**Figure 3 children-08-00480-f003:**
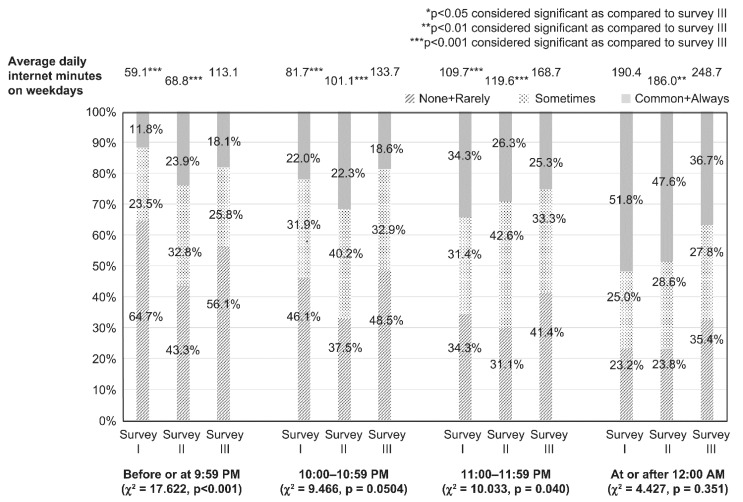
Comparison of sleepiness during classes according to bedtime on weekdays in each survey. For each bedtime on weekdays, the comparison of degree of sleepiness during classes between each survey was performed using χ^2^ analysis. For each bedtime on weekdays, a comparison of internet use duration (in minutes) on weekdays among groups as per survey year was made using ANOVA. A post hoc analysis between the group from survey III and the groups from the other surveys (survey I and II) was performed using Dunnett’s test.

**Figure 4 children-08-00480-f004:**
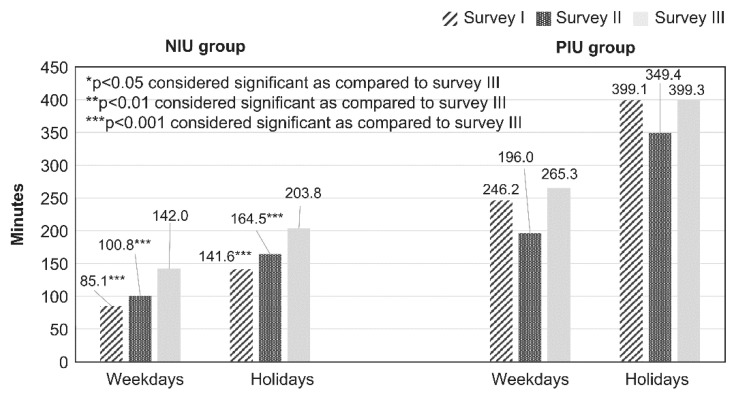
Comparison of average daily Internet minutes on weekdays and holidays for each year in the NIU and PIU groups. A comparison of the internet use duration (in minutes) on weekdays and holidays among the groups for each survey year was made using ANOVA. A post hoc analysis between the group from survey III and those from other surveys (surveys I and II) was performed using Dunnett’s test.

**Table 1 children-08-00480-t001:** Comparisons of age, sex, bedtime, sleepiness during classes, and frequency of participation in lessons or crammers between the surveys.

	Survey I—2018	Survey II—2019	Survey III—2020	Statistics
	(*N* = 734)	(*N* = 734)	(*N* = 802)	
Age				*χ*^2^ = 22.289, *p* < 0.001
12 years old	595 (81.1%)	606(82.6%)	589 (73.4%)	
13 years old	139 (18.9%)	128 (17.4%)	213 (26.6%)	
Sex				*χ*^2^ = 1.539, *p* = 0.463
Male	365 (49.7%)	352 (48.0%)	410 (51.1%)	
Female	369 (50.3%)	382 (52.0%)	392 (48.9%)	
Bedtime on weekdays				*χ*^2^ = 18.839, *p* = 0.016
Before or at 9:59 PM	170 (23.2%)	180 (24.5%)	221 (27.6%)	
10:00–10:59 PM	304 (41.4%)	301 (41.0%)	328 (40.9%)	
11:00–11:59 PM	204 (27.8%)	190 (25.9%)	174 (21.7%)	
12:00–12:59 AM	42 (5.7%)	58 (7.9%)	60 (7.5%)	
At or after 1:00 AM	14 (1.9%)	5 (0.7%)	19 (2.4%)	
Bedtime on holidays				*χ*^2^ = 13.688, *p* = 0.090
Before or at 9:59 PM	171 (23.3%)	181 (24.7%)	207 (25.8%)	
10:00–10:59 PM	264 (36.0%)	286 (39.0%)	275 (34.3%)	
11:00–11:59 PM	209 (28.5%)	178 (24.3%)	197 (24.6%)	
12:00–12:59 AM	55 (7.5%)	66 (9.0%)	85 (10.6%)	
At or after 1:00 AM	35 (4.8%)	23 (3.1%)	38 (4.7%)	
Sleepiness during classes				*χ*^2^ = 29.026, *p* < 0.001
None	91 (12.4%)	78 (10.6%)	124 (15.5%)	
Rarely	242 (33.0%)	187 (25.7%)	259 (32.3%)	
Sometimes	215 (29.3%)	279 (38.0%)	245 (30.5%)	
Common	140 (19.1%)	144 (19.6%)	132 (16.5%)	
Always	46 (6.3%)	46 (6.3%)	42 (5.2%)	
Frequency of participation in lessons or crammers				*χ*^2^ = 23.261, *p* < 0.001
None	-	134 (18.3%)	182 (22.7%)	
Less than once a week	-	97 (13.2%)	142 (17.7%)	
2 or 3 times a week	-	326 (44.4%)	339 (42.3%)	
4 or 5 times a week	-	110 (15.0%)	104 (13.0%)	
6 or 7 times a week	-	67 (9.1%)	35 (4.4%)	

Data are expressed as *n* (%).

**Table 2 children-08-00480-t002:** Comparisons of smartphone ownership, type of internet device used, type of internet service used, average duration of daily internet use, and YDQ score between the surveys.

	Survey I—2018	Survey II—2019	Survey III—2020	Statistics
	(*N* = 734)	(*N* = 734)	(*N* = 802)	
Smartphone owner	435 (59.3%)	432 (58.9%)	560 (69.8%)	*χ*^2^ = 25.774, *p* < 0.001
Type of internet device used				
Personal computer	225 (30.7%)	231 (31.5%)	257 (32.0%)	*χ*^2^ = 0.346, *p* = 0.841
Smartphone	475 (64.7%)	517 (70.4%)	613 (76.4%)	*χ*^2^ = 25.453, *p* < 0.001
Tablet	306 (41.7%)	311 (42.4%)	416 (51.9%)	*χ*^2^ = 20.321, *p* < 0.001
Portable game console	180 (24.5%)	224 (30.5%)	268 (33.4%)	*χ*^2^ = 14.980, *p* < 0.001
Stationary game console	186 (25.3%)	184 (25.1%)	228 (28.4%)	*χ*^2^ = 2.793, *p* = 0.247
Feature phone	57 (7.8%)	37 (5.0%)	15 (1.9%)	*χ*^2^ = 29.274, *p* < 0.001
Others	65 (8.9%)	69 (9.4%)	114 (14.2%)	*χ*^2^ = 13.900, *p* < 0.001
Type of internet service used				
Information and news searching	496 (67.6%)	311 (42.4%)	392 (48.9%)	*χ*^2^ = 101.285, *p* < 0.001
E-mail, chat, and internet telephone	493 (67.2%)	485 (66.1%)	567 (70.7%)	*χ*^2^ = 4.166, *p* = 0.125
Blog and internet bulletin board	61 (8.3%)	48 (6.5%)	54 (6.7%)	*χ*^2^ = 2.100, *p* = 0.350
Social networking services	129 (17.6%)	181 (24.7%)	231 (28.8%)	*χ*^2^ = 27.025, *p* < 0.001
Online game	328 (44.7%)	336 (45.8%)	400 (49.9%)	*χ*^2^ = 4.666, *p* = 0.097
Movie and music	569 (77.5%)	607 (82.7%)	676 (84.3%)	*χ*^2^ = 12.580, *p* = 0.002
Shopping and auction	98 (13.4%)	86 (11.7%)	113 (14.1%)	*χ*^2^ = 1.966, *p* = 0.374
Others	100 (13.6%)	127 (17.3%)	149 (18.6%)	*χ*^2^ = 7.235, *p* = 0.027
Average duration of daily internet use on weekdays (min)	92.5 ± 100.3	104.8 ± 97.4	147.0 ± 115.9	*F* = 57.120, *p* < 0.001
Average duration of daily internet use on holidays (min)	153.5 ± 170.3	173.3 ± 169.5	211.4 ± 176.6	*F* = 22.529, *p* < 0.001
Average YDQ score	1.5 ± 1.5	1.5 ± 1.5	1.6 ± 1.6	*F* = 1.273, *p* = 0.280
PIU group (YDQ score, ≥5)	34 (4.6%)	32 (4.4%)	42 (5.2%)	*χ*^2^ = 0.689, *p* = 0.709

Data are expressed as *n* (%) or mean ± standard deviation. YDQ: Young’s Diagnostic Questionnaire; PIU: problematic internet use.

**Table 3 children-08-00480-t003:** Comparisons of age, sex, bedtime, sleepiness during classes, and frequency of participation in lessons or crammers between the study groups.

	NIU Group	PIU Group	Statistics
	(*n* = 2162)	(*n* = 108)	
Age			*χ*^2^ = 0.079, *p* = 0.809
12 years old	1706 (78.9%)	84 (77.8%)	
13 years old	456 (21.1%)	24 (22.2%)	
Sex			*χ*^2^ = 1.583, *p* = 0.237
Male	1067 (49.4%)	60 (55.6%)	
Female	1095 (50.6%)	48 (44.4%)	
Bedtime on weekdays			*χ*^2^ = 91.954, *p* < 0.001
Before or at 9:59 PM	558 (25.8%)	13 (12.0%)	
10:00–10:59 PM	899 (41.6%)	34 (31.5%)	
11:00–11:59 PM	539 (24.9%)	29 (26.9%)	
12:00–12:59 AM	140 (6.5%)	20 (18.5%)	
At or after 1:00 AM	26 (1.2%)	12 (11.1%)	
Bedtime on holidays			*χ*^2^ = 153.804, *p* < 0.001
Before or at 9:59 PM	547 (25.3%)	12 (11.1%)	
10:00–10:59 PM	801 (37.0%)	24 (22.2%)	
11:00–11:59 PM	555 (25.7%)	29 (26.9%)	
12:00–12:59 AM	192 (8.9%)	14 (13.0%)	
At or after 1:00 AM	67 (3.1%)	29 (26.9%)	
Sleepiness during classes			*χ*^2^ = 29.272, *p* < 0.001
None	283 (13.1%)	10 (9.3%)	
Rarely	668 (30.9%)	20 (18.5%)	
Sometimes	702 (32.5%)	37 (34.3%)	
Common	393 (18.2%)	23 (21.3%)	
Always	116 (5.4%)	18 (16.7%)	
Frequency of participation in lessons or crammers (measured only in surveys II and III)			*χ*^2^ = 0.455, *p* = 0.978
None	300 (20.5%)	16 (21.6%)	
Less than once a week	226 (15.5%)	13(17.6%)	
2 or 3 times a week	634 (43.4%)	31 (41.9%)	
4 or 5 times a week	205 (14.0%)	9 (12.2%)	
6 or 7 times a week	97 (6.6%)	5 (6.8%)	

Data are expressed as *n* (%). NIU: normal internet use (Young’s Diagnostic Questionnaire score, ≤4); PIU: problematic internet use (Young’s Diagnostic Questionnaire score, ≥5).

**Table 4 children-08-00480-t004:** Comparisons of smartphone ownership, type of internet device used, type of internet service used, and average duration of daily internet use between the study groups.

	NIU Group	PIU Group	Statistics
	(*n* = 2162)	(*n* = 108)	
Smartphone owner	1348 (62.3%)	79 (73.1%)	*χ*^2^ = 5.138, *p* = 0.025
Type of internet device used			
Personal computer	670 (31.0%)	43 (39.8%)	*χ*^2^ = 3.718, *p* = 0.056
Smartphone	1520 (70.3%)	85 (78.7%)	*χ*^2^ = 3.503, *p* = 0.066
Tablet	968 (44.8%)	65 (60.2%)	*χ*^2^ = 9.852, *p* = 0.002
Portable game console	607 (28.1%)	65 (60.2%)	*χ*^2^ = 50.889, *p* < 0.001
Stationary game console	559 (25.9%)	39 (36.1%)	*χ*^2^ = 5.575, *p* = 0.025
Feature phone	102 (4.7%)	7 (6.5%)	*χ*^2^ = 0.700, *p* = 0.358
Others	231 (10.7%)	17 (15.7%)	*χ*^2^ = 2.702, *p* = 0.112
Type of internet service used			
Information and news searching	1133 (52.4%)	66 (61.1%)	*χ*^2^ = 3.128, *p* = 0.093
E-mail, chat, and internet telephone	1451 (67.1%)	94 (87.0%)	*χ*^2^ = 18.783, *p* < 0.001
Blog and internet bulletin board	134 (6.2%)	29 (26.9%)	*χ*^2^ = 65.835, *p* < 0.001
Social networking services	491 (22.7%)	50 (46.3%)	*χ*^2^ = 31.522, *p* < 0.001
Online game	986 (45.6%)	78 (72.2%)	*χ*^2^ = 29.263, *p* < 0.001
Movie and music	1749 (80.9%)	103 (95.4%)	*χ*^2^ = 14.342, *p* < 0.001
Shopping and auction	264 (12.2%)	33 (30.6%)	*χ*^2^ = 30.440, *p* < 0.001
Others	343 (15.9%)	33 (30.6%)	*χ*^2^ = 16.063, *p* < 0.001
Average duration of daily internet use on weekdays (min)	109.6 ± 100.3	238.7 ± 164.3	*t* = −8.097, *p* < 0.001
Average duration of daily internet use on holidays (min)	170.2 ± 161.3	384.5 ± 268.7	*t* = −8.216, *p* < 0.001

Data are expressed as *n* (%) or mean ± standard deviation. NIU: normal internet use (Young’s Diagnostic Questionnaire score, ≤4); PIU: problematic internet use (Young’s Diagnostic Questionnaire score, ≥5).

**Table 5 children-08-00480-t005:** Associations between dependent (PIU: YDQ ≥ 5 point) and independent (sex, age, survey years, bedtime on weekday and holidays type of internet device used, and type of internet service used) variables in the logistic regression analysis.

	Adjusted OR *	95% CI **	*p*-Value
Sex			
Male	Reference		
Female	0.876	0.545–1.409	0.586
Age			
12 years old	Reference		
13 years old	0.920	0.558–1.518	0.744
Survey years			
Survey III (2020)	Reference		
Survey II (2019)	0.905	0.582–1.614	0.969
Survey I (2018)	1.091	0.639–1.862	0.749
Bedtime on weekdays			
Before or at 9:59 PM	Reference		
10:00–10:59 PM	1.084	0.502–2.339	0.837
11:00–11:59 PM	0.980	0.408–2.358	0.964
At or after 12:00 AM	2.185	0.828–5.766	0.114
Bedtime on holidays			
Before or at 9:59 PM	Reference		
10:00–10:59 PM	1.216	0.544–2.720	0.633
11:00–11:59 PM	1.780	0.753–4.208	0.189
At or after 12:00 AM	2.949	1.150–7.567	0.024
Type of internet device used			
Personal computer			
No	Reference		
Yes	0.876	0.556–1.380	0.569
Smartphone			
No	Reference		
Yes	0.816	0.458–1.454	0.491
Tablet			
No	Reference		
Yes	1.321	0.847–2.061	0.219
Portable game console			
No	Reference		
Yes	2.652	1.678–4.191	0.001
Stationary game console			
No	Reference		
Yes	0.818	0.508–1.317	0.408
Feature phone			
No	Reference		
Yes	1.243	0.509–3.039	0.633
Others			
No	Reference		
Yes	1.000	0.548–1.824	1.000
Typeofinternetserviceused			
Informationandnewssearching			
No	Reference		
Yes	0.741	0.465–1.179	0.205
E-mail, chatandinternettelephone			
No	Reference		
Yes	2.289	1.166–4.497	0.016
Blogandinternetbulletinboard			
No	Reference		
Yes	2.217	1.263–3.888	0.006
Socialnetworkingservices			
No	Reference		
Yes	0.985	0.588–1.650	0.954
Onlinegame			
No	Reference		
Yes	1.419	0.851–2.366	0.180
Movieandmusic			
No	Reference		
Yes	2.404	0.935–6.178	0.069
Shoppingandauction			
No	Reference		
Yes	1.832	1.104–3.038	0.019
Others			
No	Reference		
Yes	1.367	0.821–2.275	0.229

* OR: odds rate; ** CI: Confidence interval.

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
