# Peer review of "Change of Internet Use and Bedtime among Junior High School Students after Long-Term School Closure Due to the Coronavirus Disease 2019 Pandemic"

_children, 2021, doi:10.3390/children8060480_

Round 1

Reviewer 1 Report

The paper gives som interesting insights on how Covid dramatically change Internet use and likely dependency. From a chronobiological perspective it would have added greatly if also rise times and sleep hours could have been covered. The phase of rhythms might have changed. From a sleep perspective we know from other studies sleep is likely to increase by Covid restrictions. These perspectives have unfortunately not been able to be addressed. 

The sentence on non-significant data in the abstract should be removed

“The rates of problematic internet use (Young’s Diagnostic 15 Questionnaire score, ³5) were higher among the participants of the 2020 survey, but not significantly 16 so, compared with the rates observed among the participants of the previous surveys”

How was age controlled for?

“The rate of 13-year-old participants in survey III was higher than the rates of such participants in the other surveys.”

We are missing figure legends in several graphs. Also explanations of y-axis is missing.

Exactly how was post-hoc tests calculated? Are p-values in Figures representing the difference in bedtime groups within each year? Difference in internet use from the Before 21.59 group? Is the lowest value always the reference? Also in Figure 4?

The sentence is difficult to understand – is there a difference across time or not?

“The average YDQ score and rate of PIU (YDQ score, ³5) increased in survey III, but these did not significantly differ between the surveys.” I suggest you stick to the decided 5% level to define significant results.

Some drawbacks need more discussion.

Sleep time should be included. Problems to control for technical development, general internet and electronic data media use in society.

Control for age lacking. During lockdown students change environment and this could add to daytime sleepiness. Did outdoor activity decrease?

Students were on lockdown and had a forced increase of Internet use. How would an increase of daytime Internet use be significant for a delayed bedtime? This is a bit difficult to understand.

Considering the decrease of daytime sleepiness in group III, this has to be better discussed, especially since this could indict better sleep in the pandemic.

The wording of the question on bedtime should be cited in the paper, do they reflect the past week or month just “normal” bedtime.

I suggest to use some regression analysis, controlling for age, to predict Internet use. The following question is difficult to grab:

“The rate of participants who were “common” or “always” sleepy during classes tended to be higher with later bedtime.” If this is a tendency, it should not be reported as a result and a p-value or other indication has to be stated.

The result section on differences between NIU and PIU is not reflected in the title of the paper. What is the rational for including the section? It would be fair in this analysis to also include day time sleepiness, it is not fully understood if PIU also poses a risk for sleep deficit (it would normally but not necessarily during lockdown). Thus the NIU/PIU question should be analyzed in view of Covid restrictions (analyze were 2019/2020 is separated from 2021).

Also, the authors have not cleared out how much the school organization has increased the likelihood of a PIU classification. Also, since Internet use gradually increases, by access to new devices etc, how may PIU use be teased out from merely the increased dependency on Internet that are forced upon individuals. To better understand PIU there is a need to also include other factors like increase of unhealthy behaviour, social avoidance, mental health status etc.

The PIU group in the sample is very small, only 5%, and I am not sure the sample then is relevant to use in such analysis. A fruitful analysis would to also include another cut-off adapted to the current sample, and to see if “mild PIU” would indicate similar sleep threats.

Reviewer 2 Report

Thank you for opportunity to review the paper „ Internet use and bedtime in junior high school students after long-term school closure due to the COVID-19 pandemic”

The article is interesting and well written, however there are few shortcomings , that need to be corrected.

  1. Were data from survey I and survey II previously published? If so, it should be cited

  2. Sleepiness should be assesed using validated scales, ie Epworth Scale

  3. 235-237 You mean daily? Please be precise

  4. The consent of Bioethics Committee is missed

  5. Brief conslusions would be usufull, which results are really novel?

  6. Have you obtain written consent from participants?

Round 2

Reviewer 1 Report

Maybe shortened school days are the prime reason for later bed times? Why was not bed times specific for school days when they had school at home asked for, this could have been logical if the aim of the study was to evaluate COVID-19. It seems this crucial question is lacking? This should be specific in the limitations section.

The following question is difficult to understand, were students really asked to summarize sleep start at non-school days (weekends) and days with extracurricular activities (schooldays). What was the rationale behind this question? “What was the average time you went to bed on the non-schooldays, including days with extracurricular activities only, in the previous 30 days?”)

Please be more precise when explaining use of statistics. “Categorical data were assessed using c2 analysis”. What comparisons were made – between what groups, years, sex, age, all possible comparisons? Also continous data analysis has to be explained.

I think you should stick to one type of time keeping, in Figure 1, 23:59 used but in Table the AM/PM system.

In my mind a figure should be able to stand for itself, see Figure 1 and 2. Thus please explain in Figure legend that a post-hoc test (Dunnett) was used using the early bed group as baseline within each year. The new text (164) has to be referred to a specific Figure.

What are the statistics in Figure 3, please explain in legend and text more in more detail. Actually, I think the new yellow sentences have by mistake been placed before start of the legend? Average daily Internet minutes show some statistics in Figure 3 – what has been calculated? It looks like the results after post-hoc, but why is Study III used as baseline? It seems a bit un-logical in a time series analysis.

Considering the decrease of daytime sleepiness in group III, this has to be better discussed, especially since this could indict better sleep in the pandemic.

Table 5 is ok but the study would be much more interesting if you in a regression analysis could predict that Covid-19 (study III) will/will not increase Internet use when controlling for background factors (age, access mobile, personal computers etc). In a similar fashion PIU could be predicted as well as sleep start. If you could predict more AI from Covid-19 it would be interesting – this is actually what you promise in the title but still not proven in regression analysis.

271 “In our study, the proportion of participants in survey III who answered “common” or “always” sleepy during classes tended to be higher than that of participants in the other surveys who did, even though their average duration of daily internet use was longer. “ When looking at Table 1 it is difficult to agree with this sentence.

344 Comparisons to “infant” Internet use should be omitted.

353 “In the 2020 survey, the rates of PIU was not significantly different from the results of previous surveys” Since this is import it would be wise to state: “However, in the 2020 survey, the rates of PIU was not significantly different from the results of previous surveys.” This more accurately reflects the take home message from the study.

Author Response

Reviewer 1

Comment #1-1

Maybe shortened school days are the prime reason for later bed times? Why was not bed times specific for school days when they had school at home asked for, this could have been logical if the aim of the study was to evaluate COVID-19. It seems this crucial question is lacking? This should be specific in the limitations section.

Response #1-1

Thank you for your insightful comments and suggestions.

On all weekdays, shorter-duration classes were held at all junior high schools during the period when the bedtime questions were asked (from June to July 2020 when survey III was conducted). The number of school days in June 2020 was not less than that in previous years. However, shortening the school time might have affected the students’ bedtime.

We have added this in the manuscript as follows:

“As school club activities and most events were restricted and school time was shortened during the period when survey III was conducted, students might have been less physically tired than in the previous prepandemic years. Furthermore, these factors might have affected the participant’s bedtime and wake-up time in survey III.” (page 12, Line 304~)

Comment #1-2

The following question is difficult to understand, were students really asked to summarize sleep start at non-school days (weekends) and days with extracurricular activities (schooldays). What was the rationale behind this question? “What was the average time you went to bed on the non-schooldays, including days with extracurricular activities only, in the previous 30 days?”)

Response #1-2

In order for the students to answer these questions accurately, they needed to keep track of their bedtime each night for a period of 1 month.

However, these questions were in the multiple-choice format, and the students were required to choose from the following options: before or at 21:59, 22:00–22:59, 23:00–23:59, 0:00–0:59, at or after 1:00. We believed that it would be possible for many students to answer these questions with accuracy.

We have added this in the revised manuscript as follows:

“Questions other than those that sought to determine the average duration of daily internet use were in the multiple-choice format.”

(Page 3, Line 111~)

Comment #1-3

Please be more precise when explaining use of statistics. “Categorical data were assessed using c2 analysis”. What comparisons were made – between what groups, years, sex, age, all possible comparisons? Also continous data analysis has to be explained.

Response #1-3

We have described our methods in detail in the statistical analysis section.

(Page 3, Line 120~)

Comment #1-4

I think you should stick to one type of time keeping, in Figure 1, 23:59 used but in Table the AM/PM system.

Response #1-4

Thank you for your suggestion.

We have revised the time format in Figures 1, 2, and 4 to the AM/PM system.

Comment #1-5

In my mind a figure should be able to stand for itself, see Figure 1 and 2. Thus please explain in Figure legend that a post-hoc test (Dunnett) was used using the early bed group as baseline within each year. The new text (164) has to be referred to a specific Figure.

Response #1-5

We have revised the statistical explanation in each figure and its legend.

Comment #1-6

What are the statistics in Figure 3, please explain in legend and text more in more detail. Actually, I think the new yellow sentences have by mistake been placed before start of the legend? Average daily Internet minutes show some statistics in Figure 3 – what has been calculated? It looks like the results after post-hoc, but why is Study III used as baseline? It seems a bit un-logical in a time series analysis.

Response #1-6

We believe that the explanation regarding the statistics in Figure 3 was insufficient.

We have revised the explanation regarding statistical analyses in Figure 3 and the corresponding figure legend.

In general, the first survey year (survey I, 2018) is set as the baseline.

However, our study objective was to investigate the life style and internet use after long-term school closure because of the COVID-19 pandemic; thus, we considered survey III as the baseline.

Comment #1-7

Considering the decrease of daytime sleepiness in group III, this has to be better discussed, especially since this could indict better sleep in the pandemic.

Response #1-7

We have included some recent literature references to discuss the decrease in daytime sleepiness in Survey III.

Comment #1-8

Table 5 is ok but the study would be much more interesting if you in a regression analysis could predict that Covid-19 (study III) will/will not increase Internet use when controlling for background factors (age, access mobile, personal computers etc). In a similar fashion PIU could be predicted as well as sleep start. If you could predict more AI from Covid-19 it would be interesting – this is actually what you promise in the title but still not proven in regression analysis.

Response #1-8

We have included bedtime on weekdays and holidays in the logistic regression analysis in Table 5.

Comment #1-9

271 “In our study, the proportion of participants in survey III who answered “common” or “always” sleepy during classes tended to be higher than that of participants in the other surveys who did, even though their average duration of daily internet use was longer. “ When looking at Table 1 it is difficult to agree with this sentence.

Response #1-9

Thank you for your important comment.

We have revised the text as follows:

“In our study, the proportion of participants in survey III who answered “common” or “always” sleepy during classes showed a tendency to be lower than that of participants in the other surveys, although their average duration of daily internet use was longer.”

(page 12 Line 300)

Comment #1-10

344 Comparisons to “infant” Internet use should be omitted.

Response #1-10

We have omitted “infant” from the sentence.

Comment #1-11

353 “In the 2020 survey, the rates of PIU was not significantly different from the results of previous surveys” Since this is import it would be wise to state: “However, in the 2020 survey, the rates of PIU was not significantly different from the results of previous surveys.” This more accurately reflects the take home message from the study.

Response #1-11

We have revised the manuscript as per your comment.

(Page 14 Line 393)
